

# 3D reconstruction of horizontal and vertical quasi-geostrophic currents in the North Atlantic Ocean

Sarah Asdar[1], Daniele Ciani[2], and Bruno Buongiorno Nardelli[1]

[1]Consiglio Nazionale delle Ricerche - Istituto di Scienze Marine (CNR-ISMAR), Naples, Italy
[2]Consiglio Nazionale delle Ricerche - Istituto di Scienze Marine (CNR-ISMAR), Rome, Italy

**Correspondence:** Sarah Asdar (sarah.asdar@na.ismar.cnr.it)

**Abstract.** In this paper we introduce a new high-resolution (1/10°) data-driven dataset of 3D ocean currents developed by the National Research Council of Italy in the framework of the European Space Agency World Ocean Circulation project: the WOC-NATL3D dataset. The product domain extends over a wide portion of the North Atlantic Ocean from the surface down to 1500 m depth , and the dataset covers the period between 2010 and 2019. To generate this product, a diabatic quasi-geostrophic diagnostic model is applied to data-driven 3D temperature and salinity fields obtained through a deep learning technique, along with ERA5 fluxes and empirical estimates of the horizontal Ekman currents based on input provided by the European Copernicus Marine Service. The assessment of WOC-NATL3D currents is performed by direct validation of the total horizontal velocities with independent drifter estimates at various depths (0, 15 and 1000 m) and by comparing them with existing reanalyses that are obtained through the assimilation of observations into ocean general circulation numerical models. Our estimates of the ageostrophic components of the flow improve the total horizontal velocity reconstruction, being more accurate and closer-to-observations than model reanalyses in the upper layers, also providing an indirect proof of the reliability of the resulting vertical velocities. The reconstructed WOC-NATL3D currents are freely available at https://www.worldocean circulation.org/Products#/metadata/0aa7daac-43e6-42f3-9f95-ef7da46bc702 (Buongiorno Nardelli, 2022).

## 1 Introduction

As a key component of Earth's climate system, the ocean plays a critical role in regulating global climate patterns. On the other way round, the marine environment is increasingly being impacted by climate change, with far-reaching consequences on various ecological processes and organisms (e.g. Poloczanska et al., 2013; Doney et al., 2012). Changes in sea surface temperatures, ocean currents, and precipitation patterns are disrupting the delicate balance of marine ecosystems, resulting in shifts in habitat distribution and migration patterns, alterations in nutrient availability, and changes in the physical and chemical properties of seawater (Constable et al., 2014; van Gennip et al., 2017; Du et al., 2019).

In this context, many uncertainties persist regarding the factors influencing fish ecology and abundance, beyond fishing pressure. These include uncertainties surrounding the change in oceanic currents, which can affect the distribution of spawning areas, the survival and dispersal of larvae, the availability of food for larvae, juveniles, and adult fishes, as well as their



migratory behaviors. Additionally, the drivers and patterns of variability and diversity among planktonic organisms, which are

essential in the marine ecosystem and directly impact higher trophic levels, remain poorly understood (Ibarbalz et al., 2019).

In many cases, high resolution data are needed to correctly account for relevant dynamical regimes, especially when transport and dispersion processes are expected to be dominant. To demonstrate how innovative approaches can contribute to address these fundamental gaps in knowledge and to enhance scientific advice for fishery management, the European Space Agency included a specific "Theme" dedicated to "Sustainable fisheries" within its World Ocean Circulation (WOC) project (www.wo

rldoceancirculation.org, last accessed on 22/07/2023). Specifically, in the framework of WOC, new methodologies have been proposed to combine satellite data, in-situ measurements, atmospheric forcings, and diagnostic models, in order to obtain high spatial resolution (mesoscale-resolving) estimates of the 3D currents (including its vertical component), to be used for specific case studies.

Actually, direct in situ measurements of ocean currents are still quite limited and, due to its small magnitude, measurements

of the vertical velocity remain one of the biggest challenges in oceanography (Tarry et al., 2021; Comby et al., 2022). Vertical velocities are therefore generally inferred indirectly, and a common approach to diagnose them is to use the quasi-geostrophic omega equation (Tintoré et al., 1991; Giordani et al., 2006; Canuto and Cheng, 2017; Qiu et al., 2020).

Within WOC, a daily 3D ocean current product has been developed, including the vertical component, at a mesoscale-resolving spatial resolution (1/10°x1/10°). This product covers a wide section of the central/North Atlantic Ocean (6°W-

76°W, 20°N-50°N). The chosen domain encompasses the Eastern Boundary Upwelling Systems along the African and Iberian coasts, which serve as important fishing grounds for several species significant for both local communities and commercial exploitation (Kämpf and Chapman, 2016). It also includes the Sargasso Sea, the exclusive location where threatened European and American eels reproduce (Dekker, 2019), and the Gulf Stream area. This entire domain holds immense importance for fishery activities and is identified as a key area within international conventions for the conservation of fishing resources,

such as tuna and tuna-like fishes under ICCAT (International Commission for the Conservation of Atlantic Tunas). Prototypal WOC-NATL3D data have been already used to investigate the role of the 3D dispersion of eels' larvae from the Sargassum towards the European coasts (Munk et al., 2023).

Building on recent work carried out in the framework of the European Copernicus Service - Multi Observation Thematic Assembly Centre to develop OMEGA3D product (Buongiorno Nardelli, 2020b), the new WOC-NATL3D product has been

obtained by solving a diabatic $\mathbf{Q}$ vector formulation of the quasi-geostrophic version of the Omega equation (Buongiorno Nardelli et al., 2018; Giordani et al., 2006), whose vertical mixing terms are estimated by combining a modified version of the K-profile parameterization (KPP; Smyth et al., 2002) and empirical values based on a simplified Ekman dynamics parameterization. Once the Omega equation is solved, horizontal ageostrophic components are also estimated.

The purpose of this paper is to provide an evaluation of 3D reconstruction of the quasi-geostrophic horizontal and vertical

currents from this new North Atlantic data-driven 3D Currents product (WOC-NATL3D hereafter). The dataset achieves a daily resolution and is computed on a 1/10°x1/10° horizontal resolution grid, over 75 unevenly spaced vertical levels (denser close to the surface), between the surface and 1500 m depth. It covers a wide part of the North Atlantic basin and spans from 2010 to 2019. This product provides as well a 3D reconstruction of daily temperature and salinity, which is not addressed hereafter as



it has been fully detailed in Buongiorno Nardelli (2020a). WOC-NATL3D is openly distributed by Ifremer/CERSAT through
the project webpage https://www.worldoceancirculation.org/Products#/metadata/0aa7daac-43e6-42f3-9f95-ef7da46bc702

The paper is organised as follow: section 2 describes the input datasets and the one used for the evaluation of the production.The Long-Short Term Memory network and 3D current methods are also defined in this section. Direct comparison of
the vertical velocity and the reconstruction performance are assessed in Section 3. Data availability is described in Section 4.
Finally, the results are discussed and summarised in Section 5.

# 2 Data and methods

## 2.1 Input datasets

### 2.1.1 Reconstructed 3D tracer fields

The density field is necessary to solve the Omega equation. Even if full details on the 3D reconstruction used within WOC-NATL3D processing chain are given in Buongiorno Nardelli (2020a), we recall here the main processing steps. A deep learning
technique based on a Long-Short Term Memory network (LSTM) is used to optimize the reconstruction of 3D temperature and
salinity fields. Then, the density, necessary to solve the Omega equation, is deduced through the standard UNESCO formula.
This 2D-to-3D processing requires 2D input fields of Sea Surface Temperature (SST), Sea surface salinity (SSS), Absolute
Dynamic Topography (ADT), vertical profiles of temperature and salinity from in situ sensors and climatological 3D fields,
described below:

**Sea Surface Temperature**. SST data are from the L4 multi-year reprocessed Operational Sea Surface Temperature and
Sea Ice Analysis (OSTIA), developed by the U.K. Met Office and distributed by the European Copernicus Marine Environment Monitoring Service (Copernicus, product ID:SST_GLO_SST_L4_REP_OBSERVATIONS_010_011). OSTIA
combines and interpolates in situ observations from HadIOD and data (ESA SST CCI, C3S, EUMETSAT and REMSS)
to provide global daily gap-free maps of foundation SST (i.e., values that are not affected by the diurnal cycle) and ice
concentration at 1/20° horizontal grid resolution. The OSTIA SST considered here covers the period 2010-2019 and was
sub-sampled to 1/10° resolution. The resulting grid is taken as the final grid used for the pre-processing of the other
surface datasets.

**Sea surface salinity**. SSS data were obtained by adapting the multidimensional optimal interpolation algorithm used
within the Copernicus Marine Service to retrieve the global SSS product (product ID: MULTIOBS_GLO_PHY_S_
SURFACE_MYNRT_015_013) to a daily processing over the 1/10° North Atlantic grid. Copernicus weekly SSS fields
(taken for from 2010 to 2019) were linearly interpolated in time between the two nearest analysis dates in order to obtain
a daily background. A cubic spline interpolation method was used to upsize the grid resolution of the weekly first guess
field from 1/4° to 1/10°. Combined satellite and in situ SSS observations were then interpolated on a daily base including
information from OSTIA SST in the computation of the weights.



**Absolute Dynamic Topography**. Using the dataset of optimal currents described in Buongiorno Nardelli and Ciani (2022) (see their section 2.4) and based on Ciani et al. (2020), a new ADT product (WOC-NATL2D) is built (see their section 2.5). Before being used as input to the LSTM model, an additional processing is performed in order to make them coherent with in situ steric heights. Based on Buongiorno Nardelli et al. (2017), the ADT is adjusted by applying a linear regression between in situ steric heights and colocated ADT data in the neighborhood of each grid point, considering matchups within a temporal window of $\pm$ 10 days.

**In situ profiles**. The vertical hydrographic profiles come from the quality controlled Argo and CTD profiles produced by Copernicus CORA 5.2 (product ID: INSITU_GLO_PHY_TS_DISCRETE_MY_013_001, Szekely et al., 2019). For this study, the period 2010-2019 was selected and data were interpolated using a spline on a vertical grid with uniform spacing (10 m intervals). Steric heights were calculated using a reference level of 1500 m.

**3D climatology**. 3D monthly climatological temperature and salinity were extracted for the period 2010-2019 from the World Ocean Atlas 2013 (Locarnini et al., 2002; Zweng et al., 2013). These climatologies are originally estimated on a $1/4° \times 1/4°$ grid. In order to compute anomaly fields from daily observations on the $1/10°$ WOC-NATL3D grid, the climatological data of the first 1500 m were first interpolated through a spline on a regularly spaced vertical (10 m intervals) and then the resolution was upsized to $1/10°$ via a cubic spline interpolation.

### 2.1.2 Surface air-sea fluxes

These fields are extracted from the ERA5 global atmospheric reanalysis produced by the European Centre For Medium-Range Weather Forecasts (ECMWF). Hersbach et al. (2020) provide a complete description of ERA5 reanalysis. This study uses the mean daily ERA5 fields of the zonal and meridional components of the turbulent surface stress, the surface latent and heat flux, the surface net solar and thermal radiation, as well as total precipitation and evaporation (needed to estimate the equivalent surface salinity flux), mapped onto the $1/10°$ WOC-NATL3D grid via cubic spline interpolation over the period 2010-2019.

### 2.1.3 Ekman currents

The modelled Ekman currents used to estimate the omega diabatic forcing term are provided by Copernicus L4 multi-year global total velocity product, which provides the velocity fields at 0 m and 15 m, with a 3 hours frequency and a spatial resolution of $1/4°$, on a regular grid (product ID: MULTIOBS_GLO_PHY_REP_015_004). These data are produced by combining Ekman currents simulated by using Rio et al. (2014)'s approach and geostrophic surface currents derived from satellite altimetry. Here, the daily averaged fields from 2010 to 2019 are used and adjusted to the high-resolution WOC-NATL3D grid via cubic spline interpolation.



## 2.2 Datasets used for comparison and evaluation

### 2.2.1 Model reanalyses

The first dataset used for comparison is the version 3.7.2 of the Simple Ocean Data Assimilation product (SODA hereafter), an ocean global reanalysis, which extends from 1980 to 2016 (Carton et al., 2018). The data were downloaded from http://www.soda.umd.edu/soda3_readme.htm (last access: 27/02/2023) for the period 2010-2016. This reanalysis is based on the Modular Ocean Model, version 5, ocean component of the Geophysical Fluid Dynamics Laboratory CM2.5 coupled model (Delworth et al., 2012), with fully interactive sea ice. This product assimilates hydrographic profiles from the World Ocean

Database (Boyer et al., 2013) and in situ and satellite SST observations. SODA provides an estimation of the vertical velocity with a horizontal resolution of 1/4° and 50 vertical levels every 5 days (5-day average).

The second model used is the global eddy-resolving ocean reanalysis product GLORYS12v1 (hereafter GLORYS) distributed by the Copernicus Marine Service (product ID: GLOBAL_MULTIYEAR_PHY_001_030), of 1/12° horizontal resolution and 50 vertical levels. It covers the period 01/01/1993 to 31/12/2020 and provides estimates of 3D daily mean currents. The model

component is NEMO forced at the surface by ECMWF ERA-Interim reanalysis and assimilates along track altimeter data (Sea Level Anomaly), SST, sea ice concentration and in situ temperature and salinity vertical profiles.

### 2.2.2 Ocean drifters/floats

The first dataset used in this study comes from the Global Drifter Program (GDP) from NOAA (Lumpkin et al., 2017). A subset of drifting buoys is selected in the North Atlantic Ocean (6°W–76°W, 20°N–50°N) for the period 2010-2019. Both undrogued

and drogued drifters are considered, providing estimates of oceanic velocity at 0 m (sea surface) and 15 m, respectively. In order to discard the inertial oscillations, a low-pass filter is applied to the 6-hourly drifter observations consisting in averaging the data over a moving time window inversely scaled with the Coriolis parameter (as in Buongiorno Nardelli et al., 2018).

The second dataset YoMaHa'07 (hereafter YOMAHA) provides estimates of deep and surface currents assessed from trajectories of Argo floats at parking level and at the sea surface (Lebedev et al., 2007). These data, covering the period 1997-

2022, are distributed by Asia-Pacific Data-Research Center/International Pacific Research Center and freely accessible at http://apdrc.soest.hawaii.edu/projects/yomaha/. Most of the data in YOMAHA are provided by the floats programmed to drift at 1000 m parking level.

### 2.2.3 Satellite altimetry

The altimeter-derived surface geostrophic velocities are distributed by the Copernicus Marine Service (product ID: SEALEVEL_

GLO_PHY_L4_MY_008_047) and were processed in the framework of the multi-satellite Data Unification and Altimeter Combination System (DUACS) project. The data are provided in delayed time with a daily temporal resolution covering the period from January 1993 to December 2020 and are provided on a global regular 1/4° grid.





### 2.3 Methods

#### 2.3.1 Long-Short Term Memory network

As mentioned previously, the algorithm used to retrieve hydrographic vertical profiles is based on a stacked LSTM neural network, which is a kind of recurrent neural network that is particularly suited to learn long-term dependencies in sequential data (such as ocean vertical profiles). The model has been detailed and fully validated in Buongiorno Nardelli, 2020a. The reconstruction technique projects satellite observations at depth by learning the end-to-end mapping from surface data (taken as "predictors" together with a few ancillary information) to observed vertical profiles (our "target"). The input includes the

anomalies of SST, SSS, adjusted ADT (detailed in section 2.1.1) as well as the latitude, the longitude, and the cyclic day (day projected on a circle), while the output vector is constructed from the in situ co-located anomaly profiles of temperature, salinity and steric height (all anomalies are computed from WOA13 climatologies). First, the network is trained by adjusting its parameters to minimize the mean squared error (loss function) between the reconstructed vertical profiles and the in situ co-located anomaly profiles. 85% of the in situ profiles are used for the so called "training phase" and the remaining 15% are saved

for the validation phase. Once the algorithm has been fitted to the training data, the test phase assesses the network performances by using independent observations. To prevent over-fitting and ensure generalization during model training and also quantify the network uncertainties, a Monte-Carlo dropout strategy is applied during model training and testing. Buongiorno Nardelli (2020a) contains a complete description of the algorithm used to reconstruct the 3D temperature and salinity from surface data. The best performance was obtained with a 2-layer stacked network, including 35 hidden units in each LSTM layer. The

LSTM code has been released under the terms of the GNU General Public Licence v3 and is available at the following address: https://github.com/bbuong/3Drec. The algorithm is applied in our region of interest over the time period 2010-2019.

#### 2.3.2 3D current retrieval

WOC-NATL3D vertical velocity fields were obtained by solving the quasi-geostrophic Omega equation's diabatic **Q** vector formulation (Giordani et al., 2006; Buongiorno Nardelli et al., 2018):

$$\nabla_h^2(N^2 w) + f^2 \frac{\partial^2 w}{\partial z^2} = \nabla_h \cdot \mathbf{Q} \tag{1}$$

with $w$ being the vertical velocity, $N^2$ the Brunt-Väisälä frequency, $f$ the Coriolis parameter and $h$ indicates the horizontal components. The **Q** vector is made up of three components defining different processes, described in the set of Eq. 2 below:

$$\mathbf{Q} = 2\mathbf{Q}_{twg} + \mathbf{Q}_{th} + \mathbf{Q}_{dm}$$

$$\mathbf{Q}_{twg} = \frac{g}{\rho_0} \left( \frac{\partial u_g}{\partial x} \frac{\partial \rho}{\partial x} + \frac{\partial v_g}{\partial x} \frac{\partial \rho}{\partial y}, \frac{\partial u_g}{\partial y} \frac{\partial \rho}{\partial x} + \frac{\partial v_g}{\partial y} \frac{\partial \rho}{\partial y} \right)$$

$$\mathbf{Q}_{dm} = \frac{f}{\rho_0} \left( \frac{\partial^2}{\partial z^2} \left[ \rho K_m \left( \frac{\partial v_g}{\partial z} + \frac{\partial v_{Ekman}}{\partial z} \right) \right], -\frac{\partial^2}{\partial z^2} \left[ \rho K_m \left( \frac{\partial u_g}{\partial z} + \frac{\partial u_{Ekman}}{\partial z} \right) \right] \right)$$



$$\mathbf{Q}_{th} = \nabla_h \left( \frac{\partial}{\partial z} \left[ K_\rho \left( N^2 + \frac{g}{\rho_0} \gamma_\rho \right) \right] \right) \tag{2}$$

where $twg$, $dm$ and $th$ represent the kinematic deformation, the turbulent momentum and the turbulent buoyancy components respectively, $\rho$ is the potential density, $g$ is the gravitational acceleration and $(u_g, v_g)$ and $(u_{Ekman}, v_{Ekman})$ represent the geostrophic and Ekman horizontal velocities. The terms $K_m$, $K_\rho$ denote the turbulent viscosity/diffusivity and $\nu_\rho$ is a non-local tracer effective gradient. These terms are computed in the OMEGA3D processing through the K-Profile parametrization, with the method described in Smyth et al. (2002) which was only modified by Buongiorno Nardelli (2020b) to handle non-

staggered non-uniform vertical grids. In order to further improve the product performances at the surface level, we removed the non-local flux of momentum from the formulation of the upper layer mixing parameterization, we further constrained the viscosity values not to exceed a consistent empirical estimate and included an empirical estimation of the Ekman shear based on the Copernicus product described in 2.1.3. Specifically, in order to introduce some more realistic ageostrophic shear in the Ekman layer, we assumed that the background Ekman velocity can be approximated through an analytical fit of the

ageostrophic currents (provided at 0 m and 15 m by Copernicus Ekman empirical reconstruction, product ID: MULTIOBS_ GLO_PHY_REP_015_004) to a compressed Ekman spiral:

$$u_{Ekman}(z) = e^{\frac{z}{D_{amp}}} \left[ u_0 cos \left( \frac{z}{D_{rot}} \right) - v_0 sin \left( \frac{z}{D_{rot}} \right) \right]$$

$$v_{Ekman}(z) = e^{\frac{z}{D_{amp}}} \left[ u_0 sin \left( \frac{z}{D_{rot}} \right) + v_0 cos \left( \frac{z}{D_{rot}} \right) \right] \tag{3}$$

where $u_0$ and $v_0$ are the components of the empirical Ekman current at 0 m, the depth z is taken as positive upward and $D_{amp}$ and $D_{rot}$ represent the Ekman depth estimates obtained from the amplitude decay and the vector rotation between 0 and 15 m depth respectively (see Roach et al., 2015).

   In addition, the viscosity in the Ekman layer, illustrated by $K_m$ in Eq.2, is also constrained by an analytical profile estimated empirically (defined in Nagai et al., 2006), with a maximum viscosity $K_{max}$ derived from the local Ekman amplitude decay

scale:

$$K_{empirical} = K_{max} \left[ 1 + tanh \left( \frac{z - D_{amp}}{\delta} \right) \right] \tag{4}$$

where $\delta$ represents the thickness of the transition layer (here set to 40 m, as in Nagai et al., 2006) and $K_{max} = \frac{f D_{amp}^2}{2}$

   Note that the removal of the nonlocal flux of momentum from the formulation of the upper layer mixing parameterization and the constraint imposed on the viscosity values allow to correct the dynamically inconsistencies eventually found in the

surface layer within the Copernicus Marine Service OMEGA3D product.





The equations are numerically solved by iteratively performing a matrix inversion on a linear system in $w$. Once vertical velocities are known, they are used to retrieve ageostrophic horizontal velocities. The numerical scheme is exactly the same as the one described in Buongiorno Nardelli (2020b), which was only adapted to the higher horizontal resolution grid.

## 3 Results

### 3.1 Comparison of vertical velocity to model reanalysis

Measuring vertical velocities in the ocean is challenging mostly because there are extremely small (order of 1-100 m day$^{-1}$). As there is a very limited number of data available, WOC-NATL3D vertical velocity is compared here to SODA model reanalysis, which provides an estimation of the vertical velocity at a 1/4° resolution. As SODA presents a temporal resolution of 5 days, WOC-NATL3D vertical velocities are averaged every 5 days in order to be consistent.

The standard deviation of vertical velocity computed over 2010-2016 (overlapping period of the 2 products) at 100 m and 1000 m for WOC-NATL3D and SODA is displayed in Fig. 1. For both products, the standard deviation of retrieved vertical velocity patterns are very similar. The domain is dominated by the intense activity associated with the Gulf Stream which shows a strong signature at 1000 m (Fig. 1a, c). Generally, WOC-NATL3D reveals more intense vertical velocities likely due to a more efficient representation of the small scale processes.

### 3.2 Horizontal velocity validation

Horizontal velocities can be inferred from the vertical integration of an equation found during the analytical derivation of Eq 1 (see Eq. 3a and 3b in Buongiorno Nardelli, 2020b). In this section, the performance of WOC-NATL3D to retrieve horizontal velocities is analysed and evaluated against DUACS, GLORYS and SODA accuracies. This is obtained by comparing the horizontal currents, at some specific depths, from those products to the data from GDP drifters, in terms of biases and root mean squared differences (RMSDs). First of all, for consistency, WOC-NATL3D and GLORYS horizontal velocities are downgraded from 1/10° and 1/12° respectively to 1/4° horizontal grid and 5-day averaged to match SODA spatial and temporal resolution. DUACS velocities are 5-day averaged as well. The time span is restricted to 2010-2016 corresponding to the overlapping period of all datasets. Then, drifters and datasets are collocated in time ($\pm$ 2 days) and space (to the nearest grid point) and horizontal velocities are vertically interpolated at drifters depths (in this paper, 0 m, 15 m and 1000 m) through a weighted average of the two closest levels (for 15 m and 1000 m depths). The matchup maps are shown in supplementary material (see Appendix A). Mean biases between GDP drifters and WOC-NATL3D, DUACS, GLORYS and SODA total horizontal velocities as well as their associated RMSDs at 0 m, computed in 2° x 2° bins, are shown in Fig.2. All mean biases appear to have a similar pattern, with largest absolute values localised in the Gulf Stream, place of intense mesoscale activity, or along the African coast where a strong upwelling occurs. Those values reveal a general underestimation of the horizontal current intensity in WOC-NATL3D, altimetry and the models (Fig. 2a, c, e, g). While WOC-NATL3D, DUACS and GLORYS show equivalent mean bias values, -5.2 cm s$^{-1}$, -6.9 cm s$^{-1}$ and -5.2 cm s$^{-1}$ respectively, SODA velocities reflect more prominent differences regarding in situ





observations with a mean bias of -11.6 cm s$^{-1}$. Note that few positive bias values are observed in the Gulf Stream. Regarding the RMSD distribution, all products display the same pattern with, as expected, the strongest values located in the Gulf Stream. WOC-NATL3D and DUACS RMSD values are very close, showing RMSD mean values of 13.9 cm s$^{-1}$ and 14.7 cm s$^{-1}$

respectively. Model reanalyses reflect the highest RMSD mean values, 15.7 cm s$^{-1}$ for GLORYS and 19.7 cm s$^{-1}$ for SODA, with some RMSDs reaching up to 57 cm s$^{-1}$ for GLORYS and 81 cm s$^{-1}$ for SODA in the Gulf Stream.

To highlight the discrepancies discussed above, differences between GLORYS RMSDs, DUACS RMSDs and WOC-NATL3D RMSDs are presented in Fig. 3. A clear improvement of WOC-NATL3D velocities is observed, reflected by positive values, with respect to GLORYS, especially along the Gulf Stream (Fig. 3b). On the top panel of Fig. 3, even though the improvement

is not as strong, WOC-NATL3D reveals a slight progress compared to the simple geostrophic estimates from satellite and confirm the lower mean RMSD of WOC-NATL3D.

Biases and RMSDs are also computed at 15 m depth between the drifters and WOC-NATL3D, GLORYS and SODA (Fig. 4). Once again, WOC-NATL3D and GLORYS patterns look similar. The currents tend to be generally underestimated despite a slight overestimation of GLORYS at some locations in the Gulf Stream. WOC-NATL3D and GLORYS reveal a mean bias of

-3.3 cm s$^{-1}$ and -1.9 cm s$^{-1}$ respectively whereas SODA appears more biased with a mean value of -8.4 cm s$^{-1}$. Note that the smaller mean bias value observed for GLORYS does not necessarily reflects a better overall behaviour than the WOC-NATL3D product. The currents overestimations seen in GLORYS along the Gulf Stream might compensate the mean bias value. RMSDs at 15 m shown in Fig. 4b,d,f, display an analogous pattern to the one at the surface (see Fig. 2), with largest values concentrated along the Gulf Stream. GLORYS RMSD and WOC-NATL3D RMSD values are also compared and the result is illustrated

in Fig. 5. This figure highlights the higher RMSD values of GLORYS compared to WOC-NATL3D, with respect to drifters. The positive values show a clear improvement of WOC-NATL3D velocities over modelled velocities especially along the Gulf Stream.

The same analysis is repeated at 1000 m depth. WOC-NATL3D, GLORYS and SODA deep currents are compared to the YOMAHA observations dataset providing horizontal currents at 1000 m depth. First of all, Fig. 6a and c brings out a general

overestimation of the currents by WOC-NATL3D and GLORYS. WOC-NATL3D occurs to be more biased than GLORYS with a mean bias of 4.9 cm s$^{-1}$ against 2.9 cm s$^{-1}$ for GLORYS. The strongest WOC-NATL3D biases are observed, once again, in the Gulf Stream but also along the Azores current (35°N), reaching a maximum of 43 cm s$^{-1}$ while GLORYS bias values do not exceed 12 cm s$^{-1}$. Conversely, SODA seems to rather underestimates velocities with sparse positive biases mainly located along the Gulf Stream (Fig. 6e). Regarding RMSDs (Fig. 6b, d, f), the largest values are clearly found along the Gulf Stream

with WOC-NATL3D displaying the highest values (mean RMSD of 9.8 cm s$^{-1}$) of all. As seen in Fig. 6a, slightly strongest WOC-NATL3D RMSDs along the latitude 35°N highlight the presence of the Azores current (Fig. 6b).

To go further, WOC-NATL3D and GLORYS RMSDs are specifically compared. Fig. 7 reveals a heterogeneous pattern. The area of the Gulf Stream is dominated by negative values suggesting that GLORYS performs better than WOC-NATL3D at 1000 m. However, some locations seem to show an improvement of WOC-NATL3D deep current.

Additionally, we assessed WOC-NATL3D horizontal velocities against OMEGA3D. In order to avoid redundancy, most of OMEGA3D figures are not shown here and the reader is invited to refer to Buongiorno Nardelli (2020b), regarding the de-





scription and validation of the OMEGA3D currents at 15 m and 1000 m depth. Yet, OMEGA3D performance at the surface (0 m) was not assessed in Buongiorno Nardelli (2020b). Therefore, biases and RMSDs between GDP drifters and OMEGA3D horizontal velocities are computed at the surface and shown in Fig. 8. Before comparing with WOC-NATL3D, it is relevant to

mention that drifters and OMEGA3D data were collocated considering their concomitant locations at times $\pm$ 3 days rather than at times $\pm$ 2 days as performed previously. This is done to account for OMEGA3D temporal resolution of 7 days (match-up map in supplementary material). Unlike WOC-NATL3D (Fig. 2a), OMEGA3D shows contrasted bias values. Overall, bias values are positive which means that OMEGA3D overestimates surface horizontal velocities, but some negative values are observed along the Gulf Stream, close to the coast, and in the northern part of the domain (Fig. 8a). Besides showing the "usual"

high values along the Gulf Stream, Fig. 8b reveals also high RMSDs in the South of the domain, between 20°N and 25°N, which is not observed in the WOC-NATL3D product (Fig. 2b).

Differences between WOC-NATL3D RMSDs and OMEGA3D RMSDs at the surface are illustrated in Fig. 9. It reveals essentially positive values, which indicates that WOC-NATL3D horizontal velocities are closer to in-situ velocities than OMEGA3D horizontal velocities. This stresses the enhancement of WOC-NATL3D at solving vertical velocities close to the surface with

respect to OMEGA3D.

### 3.3 Spectral analysis

In this last section, a spectral analysis of the horizontal velocity field at 15 m is presented in order to highlight more quantitatively the energy distribution. The power spectral densities (PSDs) zonal and meridional velocity components are estimated using Welch's method based itself on Fast Fourier Transform (Fig. 10).

The analysis focuses on the Gulf Stream region, highly energetic region. Wavenumber spectra are computed in the box shown in Fig. 10-inset (45°W-68°W, 38°N-40°N) along each latitude and then time averaged over 2010-2016 to obtain a single spectrum for every product (GLORYS, SODA, OMEGA3D and WOC-NATL3D). For a better understanding, a second x-axis representing wavelengths in kilometers is added on the top of the plots. Slopes of $k^{-3}$ and $k^{-5}$ are also indicated by dashed lines.

At low wavenumbers, all zonal velocity PSDs are in agreement, indicating a fairly equivalent representation of the largest mesoscale motion (Fig. 10a). A spectral break occurs at 400-500 km, followed by a drop in energy close to $k^{-3}$ slope for GLORYS spectrum and $k^{-5}$ for SODA spectrum, which respectively show the highest and lowest variance. WOC-NATL3D and OMEGA3D spectra present an energy reduction with a slope between $k^{-3}$ and $k^{-5}$. Up to approximately 0.7 deg$^{-1}$ (122 km), OMEGA3D and WOC-NATL3D spectra lie very close. Beyond this point, OMEGA3D experiences a significant loss in

variance, while WOC-NATL3D continues with its pattern before flattening at 4 deg$^{-1}$, its effective resolution (nominal resolution 1/10°).

Meridional velocity spectra are displayed in Fig. 10b. At large scales (<0.8 deg$^{-1}$), spectra are similar and follow a slope of $k^{-3}$ except for SODA which presents less energy and a steeper slope ($k^{-5}$). At 0.2-0.3 deg$^{-1}$, spectra drop off and follow a slope of $k^{-3}$ and $k^{-5}$ for SODA. At 0.7-0.8 deg$^{-1}$, WOC-NATL3D and OMEGA3D spectra start to decrease rapidly before

flattening between 3 and 4 deg$^{-1}$ for the former and abruptly dropping off at 2 deg$^{-1}$ for the latter. Those 2 values repre-





sent WOC-NATL3D and OMEGA3D effective resolutions. In other words, even though WOC-NATL3D nominal resolution is $1/10°$, it does not fully resolve processes at scales < $1/4°$. This is probably the consequence of the use of satellite data. In fact, filtering and interpolation are inherent in the construction of L4 SST or SSS and affect the final product. Note that OMEGA3D meridional velocity spectrum becomes even less energetic than SODA spectrum around 90 km.

## 4   Data availability

The WOC-NATL3D is freely available on the World Ocean Circulation website at https://www.worldoceancirculation.org/Products#/metadata/0aa7daac-43e6-42f3-9f95-ef7da46bc702 (Buongiorno Nardelli, 2022). The characteristics of the product are summarized in table 1.

## 5   Conclusions

In the framework of the ESA-WOC project Theme 2 "Sustainable fisheries", a new product, WOC-NATL3D, has been developed to provide a high resolution reconstruction of ocean 3D dynamics in the North Atlantic. WOC-NATL3D product is based on the omega diagnostic tool originally used in the framework of the European Copernicus Marine Service to deliver a global product of 3D ocean currents that includes the vertical component of the velocity (OMEGA3D, Buongiorno Nardelli, 2020b). The OMEGA3D product was built using a method based on the quasi-geostrophic omega equation and its horizontal velocities generally display higher accuracy than velocities provided by model reanalyses when compared to independent drifter observations. However, OMEGA3D product is more suited for studies of the global dynamics and associated interannual variability than for applications targeting an accurate assessment of mesoscale-driven dispersion and transport, such as those required by ESA-WOC project. To build this new product and improve its accuracy in the layers close to the surface, we have thus adapted the omega diagnostic tool to a high resolution grid and implemented some modifications in the formulation of the **Q** vector initially used by Buongiorno Nardelli (2020b). These especially concern the component defining the turbulent momentum. By taking advantage of the modelled horizontal Ekman currents provided by Copernicus at two different depths, a background empirical ageostrophic shear term is introduced. It is assumed that the background Ekman velocity can be approximated through an analytical fit of the ageostrophic currents to a compressed Ekman spiral. The viscosity within the Ekman layer is also empirically constrained to reduce the differences between the background and retrieved horizontal ageostrophic velocities. Furthermore, potential density and geostrophic currents, used to compute the forcing terms due to the flow deformation in the omega equation, are reconstructed using a stacked Long Short-Term Memory neural network that projects sea surface satellite data at depth after training with sparse co-located in situ vertical profiles (Buongiorno Nardelli, 2020a).

The WOC-NATL3D vertical velocity was compared to the model reanalysis SODA, one of the rare product that also provide vertical velocities. WOC-NATL3D shows more intense values and, due to its higher effective spatial resolution, is able to capture small scale dynamics. Total horizontal velocities, inferred from the vertical velocities, were assessed through an intercomparison with model reanalyses and altimeter data, based on the statistics of the differences computed against drifters data.





As the other products, close to the surface, WOC-NATL3D generally underestimates the total horizontal velocities compared to GDP drifters. Both WOC-NATL3D and GLORYS display a lower bias compared to satellite or SODA, with WOC-NATL3D achieving the lowest RMSD values. At 15 m, the systematic error in WOC-NATL3D, estimated with respect to GDP, is only slightly higher than in GLORYS. At 1000 m, taking YOMAHA drifter velocity estimates as reference, WOC-NATL3D reflects the highest biases and RMSD values, and a direct comparison between WOC-NATL3D RMSDs and GLORYS RMSDs highlights a deterioration of the horizontal currents in WOC-NATL3D. Comparing directly WOC-NATL3D to OMEGA3D confirms an improved performance of the new empirical parameterization in the upper layer. Spectral analysis evidences that WOC-NATL3D presents energy at small scales, more than SODA or OMEGA3D.

WOC-NATL3D behaves well but there are still some caveats to this data-driven approach. As for OMEGA3D, WOC-NATL3D is not designed for studying coastal dynamics (due to Dirichelet conditions imposed at topographical boundaries, see Buongiorno Nardelli, 2020b). Despite this potential limitation, WOC-NATL3D has demonstrated a good performance in the upper layer of the ocean with horizontal velocities and so of vertical velocities. To conclude, this work aims to contribute to studies of the North Atlantic region by offering a new gridded product of the data-driven 3D reconstruction of ocean currents at high resolution.

**Appendix A:  Matchup database**

In order to built the matchups database, drifters (GDP and YOMAHA) and datasets (WOC-NATL3D, DUACS, GLORYS, SODA) are collocated in space and time. First, only drifters data with a date common to all datasets (according to the chosen depth) are kept. Then, data are spatially collocated by simply finding the closest grid point in the products. Horizontal velocities are vertically interpolated at drifters depths through a weighted average of the two closest levels. Fig. A1 illustrates the number of matchups between WOC-NATL3D, DUACS, GLORYS, SODA and GDP drifters at 0 m and 15 m (note that DUACS data are not taking in account to build matchups at 15 m) and Fig. B1 shows the number of matchups between WOC-NATL3D, GLORYS, SODA and YOMAHA drifters at 1000 m. In this paper, 41820 drifters at the surface, 35443 at 15 m and 2036 at 1000 m are used to performed the analyses over the period 2010-2016.

*Author contributions.*  BBN developed the WOC-NATL3D product. DC developed the WOC-NATL2D product. SA carried out the inter-comparison and validation analyses. SA wrote the manuscript. BBN and DC contributed to the writing and revision of the text and figures

*Competing interests.*  The authors declare that there is no conflict of interest.

*Financial support.*  This work has been carried out as part of the ESA World Ocean Circulation project (ESA Contract No. 4000130730/20/I-NB)





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

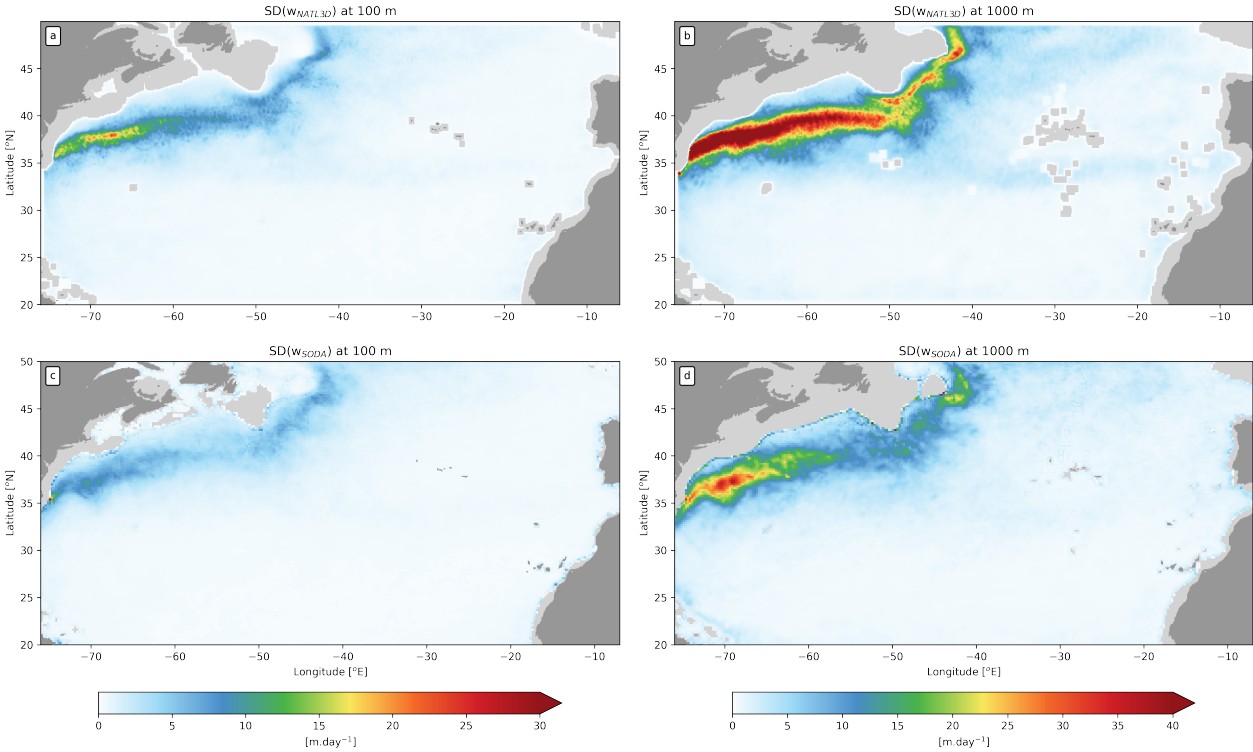

**Figure 1.** Vertical velocity standard deviation (m day$^{-1}$) for WOC-NATL3D (top) and SODA (bottom) at 100 m (a,c) and 1000 m (b,d). The standard deviation is computed over the period 2010-2016.



**Figure 2.** Mean biases (left panels) and RMSDs (right panels) between GDP drifters and WOC-NATL3D (a,b), DUACS (c,d), GLORYS (e,f) and SODA (g,h) surface currents in 2°x 2° bins, computed over the period 2010-2016.



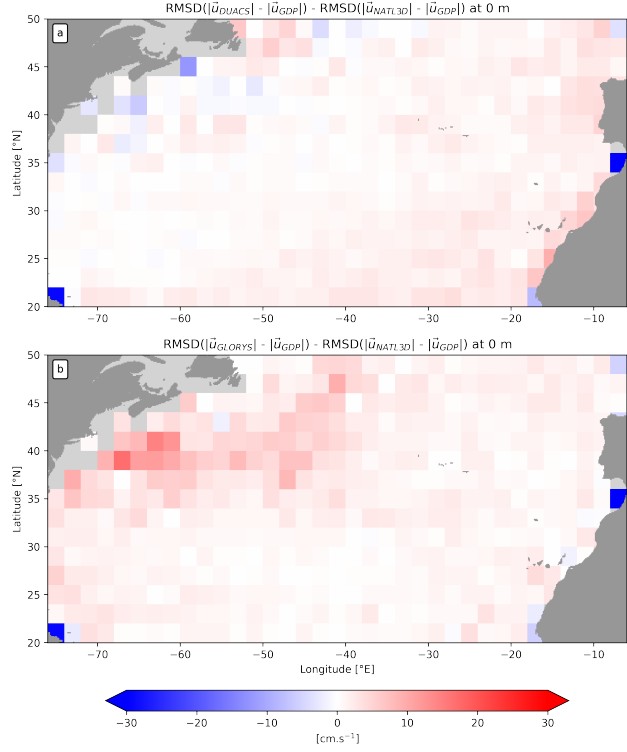

**Figure 3.** The top panel is the difference between WOC-NATL3D RMSD (Fig. 2b) and DUACS RMSD (Fig. 2c) and the bottom panel is the difference between WOC-NATL3D RMSD (Fig. 2b) and GLORYS (Fig. 2d). They are computed at the surface over the period 2010-2016. The RMSDs are computed from GDP drifter velocities.







**Figure 4.** Mean biases (left panels) and RMSD (right panels) between GDP drifters and WOC-NATL3D (a,b), DUACS (c,d), GLORYS (e,f) and SODA (g,h) currents at 15 m depth in 2° x 2° bins, computed over the period 2010-2016.



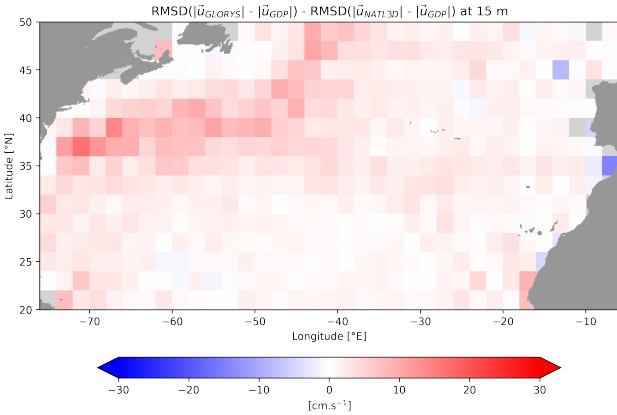

**Figure 5.** Difference between WOC-NATL3D RMSD (Fig. 4b) and GLORYS (Fig. 4d) at 15 m depth, over the period 2010-2016. The RMSDs are computed from GDP drifter velocities.







**Figure 6.** Mean biases (left panels) and RMSDs (right panels) between YOMAHA drifters and WOC-NATL3D (a,b), GLORYS (c,d) and SODA (e,f) currents at 1000 m in $2°\text{x}2°$ bins, computed over the period 2010-2016.





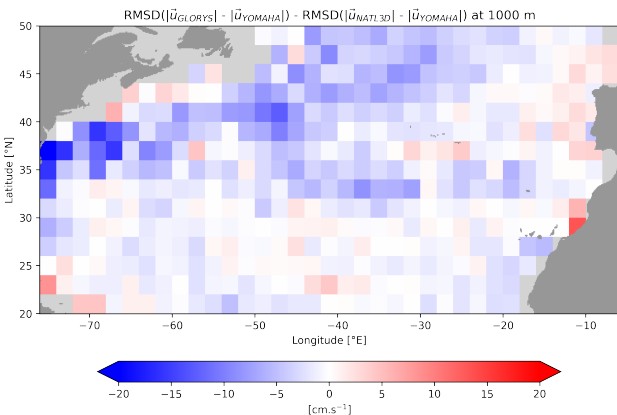

**Figure 7.** Difference between WOC-NATL3D RMSD (Fig. 6b) and GLORYS RMSD (Fig. 6d) at 1000 m depth, over the period 2010-2016. The RMSDs are computed from YOMAHA drifter velocities.





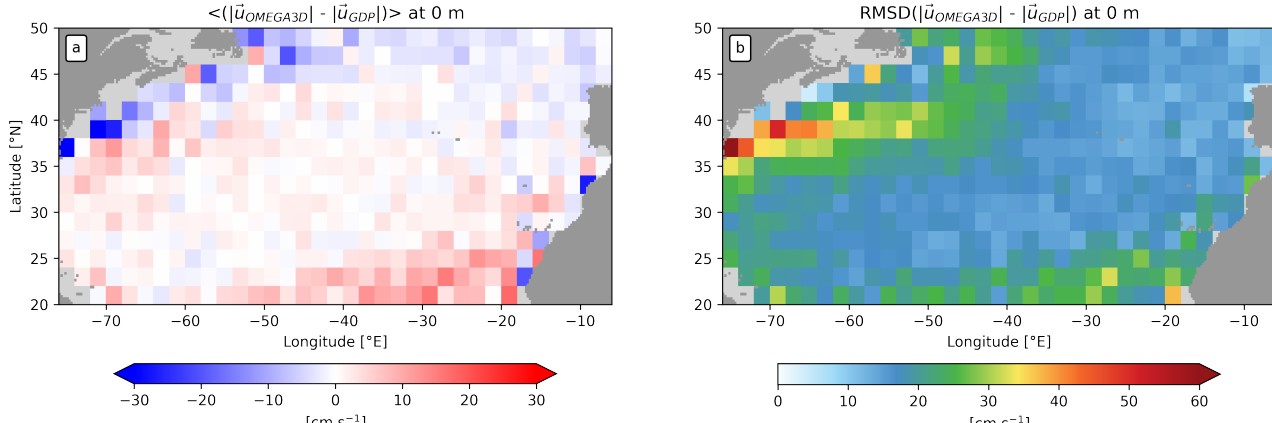

**Figure 8.** Mean bias (a) and RMSD (b) between GDP drifters and OMEGA3D surface currents in 2°x2° bins, computed over the period 2010-2016.





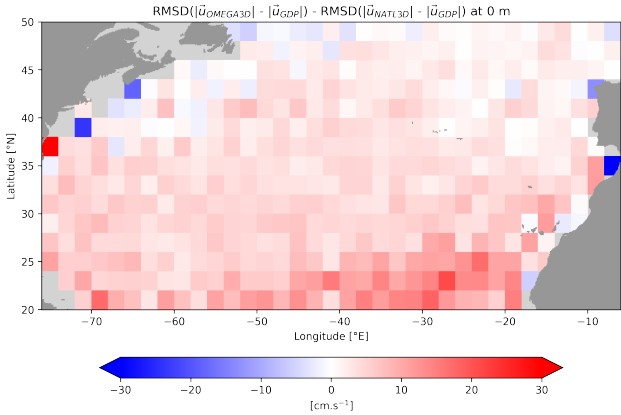

**Figure 9.** Difference between WOC-NATL3D RMSD (Fig. 2b) and OMEGA3D RMSD (Fig. 8b) at the surface, over the period 2010-2016. The RMSDs are computed from GDP drifter velocities.



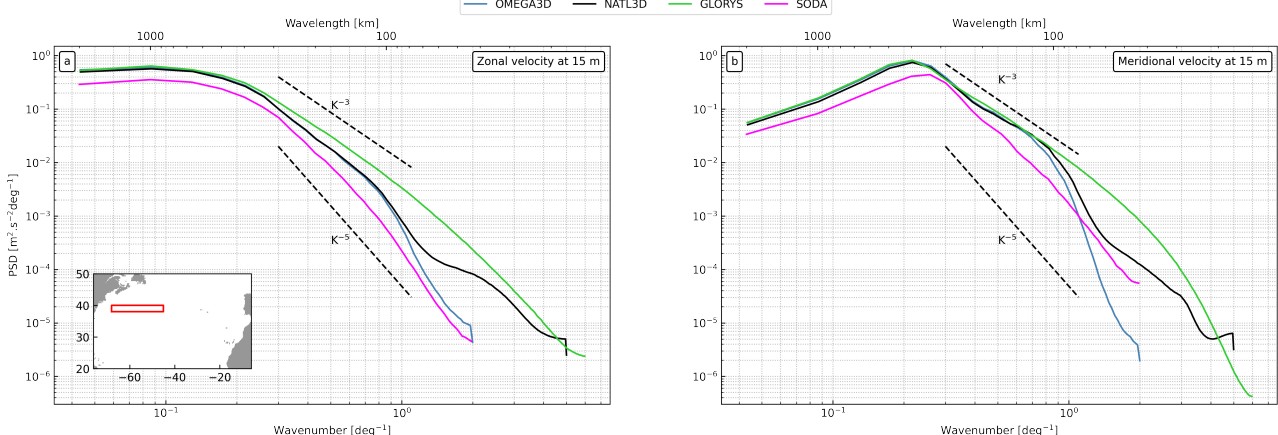

**Figure 10.** Power spectra density ($m^2 s^{-2} deg^{-1}$) of horizontal velocity at 15 m depth averaged over 2010-2016: zonal component (a) and meridional component (b). The spectra are performed in the box shown in the inset map. Dashed lines indicate slopes of -3 and -5.



**Table 1.** Description of the product

| | |
|---|---|
| WOC product ID | WOC-L4-CUR-WOC-NATL3D_REP-1D |
| Geographical coverage | Central/North Atlantic<br>6°W-76°W, 20°N-50°N |
| Spatial resolution | 0.1° on a regular grid<br>75 vertical layers<br>Depth range: 1.25-1482.50 m |
| Temporal coverage | From 1 January 2010 to 31 December 2019 |
| Temporal resolution | Daily fields |
| Variables | uo (m s$^{-1}$): eastward velocity<br>vo (m s$^{-1}$): northward velocity<br>wo (m day$^{-1}$): vertical velocity<br>to (K): temperature<br>so (PSU): salinity |
| Format | NetCDF |



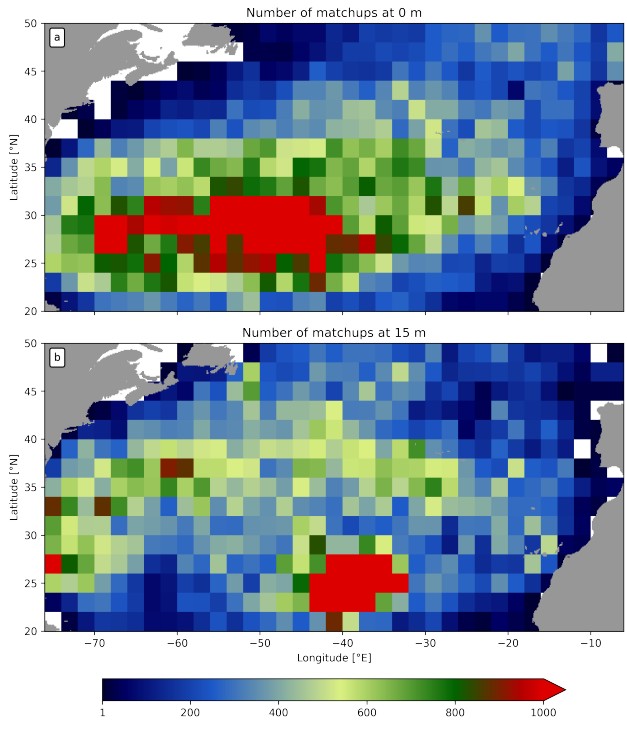

**Figure A1.** Number of matchups between WOC-NATL3D, DUACS, GLORYS, SODA and GDP drifters at 0 m (a) and between WOC-NATL3D, GLORYS, SODA and GDP drifters at 15 m (b). These matchups numbers are computed computed over the period 2010-2016 and within 2°x2° bins.

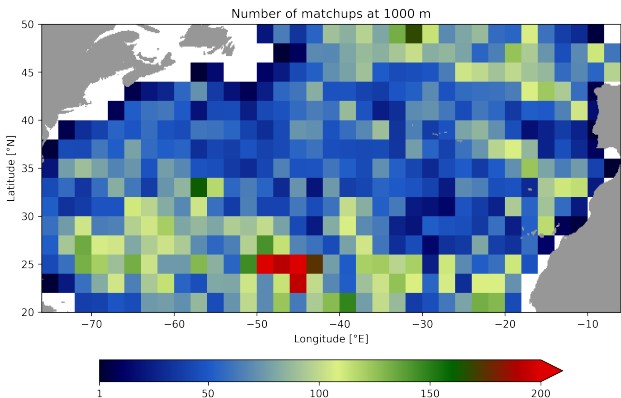

**Figure B1.** Number of matchups between WOC-NATL3D, GLORYS, SODA and YOMAHA drifters at 1000 m, computed computed over the period 2010-2016 and within 2°x2° bins.



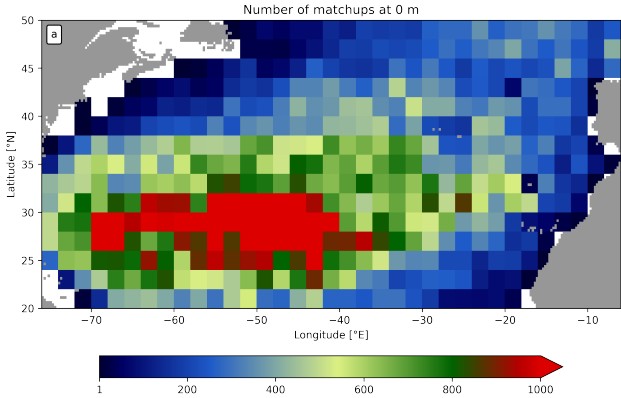

**Figure C1.** Number of matchups between WOC-NATL3D, OMEGA3D and GDP drifters at 0 m, computed computed over the period 2010-2016 and within 2°x2° bins.