# Peer review of "3D reconstruction of horizontal and vertical quasi-geostrophic currents in the North Atlantic Ocean"

_Earth System Science Data, 2023_

## Referee Comment (RC1)

**General comments :**

This document describes the 3D ocean current dataset developed in the framework of the World Ocean Circulation project (ESA funded) and named WOC-NATL3D.

The dataset extends over the North Atlantic, from the surface to 1500 m depth at a 1/10° spatial resolution on a daily basis. The dataset contains horizontal and vertical velocity, as well as temperature and salinity fields from 2010 to 2019.

A deep learning algorithm is used to compute 3D T/S fields which then feed a diabatic quasi-geostrophic equation together with ERA5 fluxes and horizontal Ekman currents estimates from European Copernicus Marine Service.

This product has been developed to fit the purpose of fishery applications that need higher spatial resolution vertical velocity.

This article is well written, pleasant to read and well structured. However, we are still left wanting more information about validation. It does not clearly answer the question of the validity of the WOC-NATL3D vertical velocity:

- Some figures request more discussion in the text. The authors referred to publications describing the method, but more details could have been provided in the text for a better comprehension.
- All the validation is performed on current intensity. There is no indication on the validation of direction of the current, neither on WOC-NATL3D nor the other reanalysis dataset. Vertical velocity is related to divergence of the horizontal flow but no validation of the zonal nor meridional component is shown here. Can the authors provide some justification for this choice or provide some more validation results?

**Specific comments:**

1. **Lines 30 to 35**: "*in order to obtain high spatial resolution (mesoscale-resolving) estimates of the 3D currents (including its vertical component), to be used for specific case studies.*"

The only mention of case studies in this document is between **lines 45 and 48**, referring to Munk et al, 2023 publication. Can the authors say few words about this, even if this is not the main purpose of this article?

2. **Lines 57 to 58:** "*It covers a wide part of the North Atlantic basin and spans from 2010 to 2019.*"

Can the authors say a few works to justify the choice of this time period?

3. **Line 81**: How is OSTIA dataset subsampled? Is it one point out of 2?
4. **Lines 83 to 89:** Can you add a reference to "*the multidimensional optimal interpolation algorithm used within the Copernicus Marine Service to retrieve the global SSS product*"?
5. **Lines 90 to 95:** Can you make a comment on the temporal resolution/ temporal smoothing resulting from the ± 10 days temporal window used to select the insitu profils in the ADT estimation?
6. **Lines 100 to 104:** I understand that the 3D climatology is vertically interpolated on regular 10m vertical levels. It does not match the WOC-NATL3D vertical grid (unregular spaced vertical grid). Can you provide more details?

7. **2.1.3**: about the Ekman product reference, the last reference is Mulet et al, 2021 (but based on Rio et al, 2014):

Mulet, S., Rio, M.-H., Etienne, H., Artana, C., Cancet, M., Dibarboure, G., Feng, H., Husson, R., Picot, N., Provost, C., and Strub, P. T.: The new CNES-CLS18 global mean dynamic topography, Ocean Sci., 17, 789–808, https://doi.org/10.5194/os-17-789-2021, 2021.

In the following, it seems that you only use the Ekman current from the daily MULTIOBS_GLO_PHY_REP_015_004 product (which is the sum of geostrophy+ Ekman). Can you just say a few words on how you recover the "Ekman only" component?

8. **2.1.1 to 2.1.3:** can you say a few words on why you choose a cubic spline interpolation for the different input datasets?
9. **2.2.2:** can you precise the frequency of the YoMaHa database? I think you don't use the surface YoMaHa observations in the following. Can you explain why?
10. **Lines 203 to 205:** can the authors provide some details on the dynamical inconsistencies between OMEGA3D and WOC-NATL3D and why this formulation provide a correction?
11. **3.1.** why do you choose 100 m and 1000 m levels to show SODA and WOC-NATL3D vertical velocities? Is it a good think to find more intense vertical velocity than SODA?
12. **Lines 232 to 235 on Figure 2**: we can see some spike in WOC-NATL3D statistics (Gibraltar and lower left corner) on both bias and RMSD. Can you explain? Do you apply a threshold on the minimum number of matchup into bins to compute statistics?
13. **Figure 3**: you have to qualify results of Figure 3. Because, one result is that WOC-NATL3D is closer to DUACS than to GLORYS. Is it expected as DUACS is supposed to be "only" a geostrophic current? I suppose RMSD are the same as in Figure 2, computed after a 5-days averaging (smoother data indeed.) That's only part of the signal spectrum, and this is probably why, surprisingly, DUACS better matches the drifters than GLORYS. I think, results from Figure 1 and Figure 2 need a deeper analysis and comment in **§3.2**.
14. **Lines 246 to 257: Figure 4 and 15m statistics**: it could be interesting to also add DUACS in the 15m comparison. Can you explain why bias and RMSD are lower at 15m compared to the surface? It could have been interesting to provide statistics on zonal and merdionial components separately to assess the Ekman spiral from the surface to 15m depth.
15. **Lines 258 to 270**: validation at 1000m depth. Can you provide the mean bias value of SODA? From **Figure 6**, it seems that SODA provide the lower RMSD and bias of the 3 simulations. This is not commented here. Can you complete this analysis?
16. **Lines 284 to 285**: once again, the authors only discussed on current intensity. So, can it be relevant for vertical velocity assessment?
17. **3.3 Spectral analysis:** conclusion of this section is that the effective resolution of WOC-NATL3D in the Gulf Stream area is near ¼°. Can you further comment this result? Is it enough to fit the purpose of the project and the user cases?
18. **Conclusion: Lines 328 to 330:** Can the authors say some few words about the differences between background and retrieved horizonal ageostrophic velocity? Here again, validation diagnostics on zonal/meridional components would have been useful for the discussion.

**Technical comments:**

Repetition of word "computed" in Figure A1 caption.

---

## Referee Comment (RC2)

**Review of "3D reconstruction of horizontal and vertical quasi-geostrophic currents in the North Atlantic Ocean" by Sarah Asdar, Daniele Ciani , and Bruno Buongiorno Nardelli**

The authors present a 1/10° data-driven data set of 3D ocean currents, as well as of temperature and salinity in the upper 1500 meters of the North Atlantic subtropical gyre between 20°N and 50°N, WOC-NATL3D. The data set covers the period from 2010 to 2019 with daily resolution.

The product is based on a diagnostic tool originally developed for a global product (OMEGA3D) by Bruno Buongiorno Nardelli (2022). The method is based on the the quasi-geostrophic omega equation. A deep learning technique is used to obtain the fields from Argo profiles, altimetry, SST and SSS. Also used are ERA5 air-sea fluxes and modelled Ekman currents from Copernicus.

Both products, WOC-NATL3D and OMEGA3D, are supposed to better reproduce drifter observations when compared to reanalysis products. WOC-NATL3D aims to improve accuracy near the surface, in particular by using the modelled Ekman currents. Two reanalysis products (SODA and GLORYS) as well as drifter and altimetry data are used for evaluation.

The article is written well and comprehensibly and also well structured.

Comments:

- Evaluation of the vertical velocities (section 3.1) is quite limited. I find it understandable that no comparisons with direct measurements are possible. However, an estimation of the uncertainty of the vertical velocities is desirable.
- I understand that the SODA data set was selected for comparison because vertical velocities are rare in reanalysis products. With GLORYS a second reanalysis product was selected for comparison, is there a justification for this choice?
- The labels and titles of the figures are in a small font size. The subscript letters in the titles in particular are difficult to read on a printout.

l 15 "On the other way round", I would remove this
l 108 the surface latent and "sensible" heat flux ?
l 179 I can't find an explanation of the meaning of the variable $\gamma_\rho$ in Eqn. 2
l 182 I can't find in which equation the variable $v_\rho$ is used
l 185 I would start a new paragraph, before "In order to further improve …", as the following text focuses on extensions of OMEGA3D
l 218 "… likely due …", can this be explained further ?

Figs. A1, B1, C1 "computed computed" in the caption

---

## Author Comment (AC1)

**Reviewer #1**

**General comments**

This document describes the 3D ocean current dataset developed in the framework of the World Ocean Circulation project (ESA funded) and named WOC-NATL3D.

The dataset extends over the North Atlantic, from the surface to 1500 m depth at a 1/10° spatial resolution on a daily basis. The dataset contains horizontal and vertical velocity, as well as temperature and salinity fields from 2010 to 2019.

A deep learning algorithm is used to compute 3D T/S fields which then feed a diabatic quasi-geostrophic equation together with ERA5 fluxes and horizontal Ekman currents estimates from European Copernicus Marine Service.

This product has been developed to fit the purpose of fishery applications that need higher spatial resolution vertical velocity.

This article is well written, pleasant to read and well structured. However, we are still left wanting more information about validation. It does not clearly answer the question of the validity of the WOC-NATL3D vertical velocity:

- Some figures request more discussion in the text. The authors referred to publications describing the method, but more details could have been provided in the text for a better comprehension.
- All the validation is performed on current intensity. There is no indication on the validation of direction of the current, neither on WOC-NATL3D nor the other reanalysis dataset. Vertical velocity is related to divergence of the horizontal flow but no validation of the zonal nor meridional component is shown here. Can the authors provide some justification for this choice or provide some more validation results?

We would like to thank the reviewer for taking the time to carefully review our paper. Please find below our responses and changes made to the manuscript. All the line numbers referred to here are those appearing in the track-change version of the manuscript.

**Response**: We thank you for the suggestions and do agree that the validation of the zonal and meridional components would provide additional information. As such, we have now plotted (see below) and discussed also the RMSDs of u and v (separately) at 0 m, 15 m and 1000 m. The figures have been added to the manuscript and the reviewer will find them as figures 3, 6 and 9. Some descriptions of those figures have been added:

[revised manuscript text omitted]

**Specific comments**

**1) Lines 30 to 35**: "*in order to obtain high spatial resolution (mesoscale-resolving) estimates of the 3D currents (including its vertical component), to be used for specific case studies.*" The only mention of case studies in this document is between **lines 45 and 48**, referring to Munk et al, 2023 publication. Can the authors say few words about this, even if this is not the main purpose of this article?

**Response:** We have added the following text (after the citation of Munk et al., 2023) lines 47-52: "Specifically, Lagrangian drift trajectories were simulated starting from our data-driven high-resolution reconstruction of the 3D flow, based on the eels' larvae data collected during targeted surveys. Focusing on the effects of mesoscale processes, Munk et al. (2023) found that, while eels' spawning area is delimited by temperature and salinity fronts, their dispersion patterns are mostly influenced by current shear and eddy strain, with a significant dispersal towards the North East. This result is supported by historical data, challenging common interpretations that assume a dominant initial westward advection of the entire population toward the Gulf Stream. "

**2) Lines 57 to 58:** "*It covers a wide part of the North Atlantic basin and spans from 2010 to 2019.*" Can the authors say a few works to justify the choice of this time period?

**Response:** The time period was defined following the requirements/constraints posed by the ESA World Ocean Circulation project. Specifically, the 2010 start date was chosen considering the launch of SMOS satellite and subsequent availability of space-borne surface

salinity data. The reconstruction of the 3D tracer fields (described in Buongiorno Nardelli, 2020) relied on various inputs, including these SMOS satellite data, and temporal limitation of our WOC-NATL3D product that was inherited by the available 3D tracer reconstruction. We clarified this in the text (l. 63-64): "This time period was defined following the specifications of the ESA World Ocean Circulation project, with particular consideration given to the launch of the SMOS satellite in 2010."

**3) Line 81**: How is OSTIA dataset subsampled? Is it one point out of 2?
**Response:** Yes, we use this simple method to subsample SSS from OSTIA. We clarified this in the text (l. 88): "[...], by simply selecting one point out of two."

**4) Lines 83 to 89:** Can you add a reference to "*the multidimensional optimal interpolation algorithm used within the Copernicus Marine Service to retrieve the global SSS product*"?
**Response:** Thank you for the suggestion.
Additional information has been added at lines 93-94: "The technique is able to increase the effective resolution of the interpolated SSS by taking advantage of its covariance with local SST patterns in the open ocean."
And the three following references at line 91:

- Buongiorno Nardelli, B., R. Droghei, and R. Santoleri (2016), Multi-dimensional interpolation of SMOS sea surface salinity with surface temperature and in situ salinity data, Remote Sens. Environ., doi:10.1016/j.rse.2015.12.052.;
- Droghei, R., Buongiorno Nardelli, B., and Santoleri, R. (2016). Combining In-situ and Satellite Observations to Retrieve Salinity and Density at the Ocean Surface, J. Atmos. Ocean. Tech., doi:10.1175/JTECH-D-15-0194.1;
- Sammartino, Michela, Salvatore Aronica, Rosalia Santoleri, and Bruno Buongiorno Nardelli. (2022). Retrieving Mediterranean Sea Surface Salinity Distribution and Interannual Trends from Multi-Sensor Satellite and In Situ Data, *Remote Sensing* 14, 2502: https://doi.org/10.3390/rs14102502";

**5) Lines 90 to 95:** Can you make a comment on the temporal resolution/ temporal smoothing resulting from the ± 10 days temporal window used to select the insitu profiles in the ADT estimation?
**Response:** This technique is actually fully documented in the cited references. Anyway, we would not consider the temporal window to be actually "smoothing" the correction, at least not with respect to the other approaches adopted for example within Copernicus operational framework (e.g.: Guinehut et al. (2012) rely on climatological corrections). The choice of taking that time interval was mainly driven by in situ data coverage limitations.

- Guinehut, S., Dhomps, A.-L., Larnicol, G., & le Traon, P.-Y. (2012). High resolution 3-D temperature and salinity fields derived from in situ and satellite observations. *Ocean Sci*, *8*, 845–857. https://doi.org/10.5194/os-8-845-2012

**6) Lines 100 to 104:** I understand that the 3D climatology is vertically interpolated on regular 10m vertical levels. It does not match the WOC-NATL3D vertical grid (unregular spaced vertical grid). Can you provide more details?
**Response:** The 3D climatological fields extracted from WOA13 were used to transform daily observations (SST, SSS and in situ T/S profiles) into anomaly fields, before being used for the

reconstruction of 3D temperature and salinity fields within the neural network. In order to compute these anomalies, both the observations and climatological fields were interpolated onto a uniform vertical grid with 10 m intervals. All interpolations were carried out using cubic splines. Adjustments have been made to make it clearer lines 113-115: "These climatologies are used to convert all daily observations (in situ profiles, SST and SSS) to anomaly fields. These anomaly fields are then employed in the reconstruction of 3D temperature and salinity fields within the LSTM network."

**7) 2.1.3**: about the Ekman product reference, the last reference is Mulet et al, 2021 (but based on Rio et al, 2014): Mulet, S., Rio, M.-H., Etienne, H., Artana, C., Cancet, M., Dibarboure, G., Feng, H., Husson, R., Picot, N., Provost, C., and Strub, P. T.: The new CNES-CLS18 global mean dynamic topography, Ocean Sci., 17, 789–808, https://doi.org/10.5194/os-17-789-2021, 2021.

In the following, it seems that you only use the Ekman current from the daily MULTIOBS_GLO_PHY_REP_015_004 product (which is the sum of geostrophy+ Ekman). Can you just say a few words on how you recover the "Ekman only" component?

**Response:** Thank you for this remark. Indeed, we only extract the Ekman component from the daily MULTIOBS_GLO_PHY_REP_015_004 product. This involves subtracting the geostrophic component, sourced from the Copernicus SEALEVEL_GLO_PHY_L4_MY_008_047 product, from the total velocity, and it is now mentioned in the manuscript (l. 128-129): "The Ekman component is extracted by removing the geostrophic component provided by the SEALEVEL_GLO_PHY_ L4_MY_008_047 product from Copernicus."

**8) 2.1.1 to 2.1.3:** can you say a few words on why you choose a cubic spline interpolation for the different input datasets?

**Response:** Buongiorno et al. (2012) tested different methods of interpolation, such as the simple bilinear interpolation or the Akima spline, in order to avoid introducing spurious signals in the computation of higher order derivatives through finite differences. They finally chose a classical spline method (cubic spline). Based on this paper, we decided to use the same interpolation method. The reference has been added to line 96.

- Buongiorno Nardelli, B., Guinehut, S., Pascual, A., Drillet, Y., Ruiz, S., & Mulet, S. (2012). Towards high resolution mapping of 3-D mesoscale dynamics from observations. *Ocean Science*, *8*(5), 885–901. https://doi.org/10.5194/OS-8-885-2012

**9) 2.2.2:** can you precise the frequency of the YoMaHa database? I think you don't use the surface YoMaHa observations in the following. Can you explain why?

**Response:** Thank you for the comment. Indeed, we opt not to use the YoMaHa database at the surface, as we already have data from the Global Drifter Program. The significant advantage of the latter lies in its inclusion of all drifting buoys (both drogued and undrogued), in contrast to YoMaHa, which is limited to Argo profiles, whose design is not optimized to estimate surface currents. To avoid any confusion, we have made some modifications to the sentence, at lines 151-152: "The second dataset YoMaHa'07 (hereafter

YOMAHA) provides estimates of deep and surface currents assessed from trajec- tories of Argo floats at parking level and at the sea surface (Lebedev et al., 2007)."

Regarding the frequency, YoMaHa velocities are estimated by measuring the displacement of profiling Argo floats during their submerged phase of the profiler cycle (Lebedev et al., 2007). Argo floats drift at a predefined parking pressure and surface only for data transmission through ARGOS–IRIDIUM satellites. The majority of these instruments follow a profiling cycle of approximately 10 days, with their parking level set to 1000 m. These details have been added at line 155: "[...] and follow a profiling cycle of approximately 10 days."

**10) Lines 203 to 205:** can the authors provide some details on the dynamical inconsistencies between OMEGA3D and WOC-NATL3D and why this formulation provides a correction?
**Response:** OMEGA3D data have revealed low performance in retrieving wind-driven currents at the very surface (namely in the first 2 layers). This occasionally led to higher RMSDs with respect to drifters than simple geostrophy. In fact, we realized that while the non-local KPP terms included in OMEGA3D processing contributed to obtain sufficiently accurate estimates below the surface (as shown by the validation with 15 m drifter data), not properly accounting for the vertical shear of the Ekman components (which was formerly not included) would have required considering a time-evolving solution (to allow for a full adjustment of the surface ageostrophic currents to the wind stress), so that QG Omega would not have been applicable anymore.
In fact, as shown in Buongiorno Nardelli et al., JGR2017), this contribution to the Q forcing vector in the primitive equation approximation becomes non-negligible if the Ekman shear is not properly introduced:

$$\boldsymbol{Q}_{dr} = f\left(\frac{D}{Dt}\left(\frac{\partial v_a}{\partial z}\right), -\frac{D}{Dt}\left(\frac{\partial u_a}{\partial z}\right)\right)$$

This is why we speak about "dynamical" inconsistency. The proposed modifications provide an empirical adjustment that reduces this issue.

**11) 3.1.** why do you choose 100 m and 1000 m levels to show SODA and WOC-NATL3D vertical velocities? Is it a good think to find more intense vertical velocity than SODA?
**Response:** We decided to illustrate vertical velocity at two dynamically distinct levels:
- 100 m: upper ocean (driven by both Ekman and mesoscale processes, impacting the exchanges at the base of the euphotic layer)
- 1000 m: ocean interior (dominated by internal processes).

The higher variability observed in WOC-NATL3D compared to SODA likely indicates higher effective resolution of mesoscale processes in our product, which would be a positive outcome.

**12) Lines 232 to 235 on Figure 2**: we can see some spike in WOC-NATL3D statistics (Gibraltar and lower left corner) on both bias and RMSD. Can you explain? Do you apply a threshold on the minimum number of matchup into bins to compute statistics?
**Response:** We did not apply a threshold. After verifications, it appears that some outliers (likely due to the presence of the coastline) were not correctly flagged at those locations, resulting in some spikes in the statistics. We added some details at lines 255-258: "It should

be noted that some spikes in WOC-NATL3D statistics are visible at the Strait of Gibraltar and in the lower left corner of the domain. Some aberrant velocities, likely due to the presence of a coastline, were not flagged during the computation of statistics, resulting in unusually high values of RMSD and bias."

**13) Figure 3**: you have to qualify results of Figure 3. Because, one result is that WOC-NATL3D is closer to DUACS than to GLORYS. Is it expected as DUACS is supposed to be "only" a geostrophic current? I suppose RMSD are the same as in Figure 2, computed after a 5-days averaging (smoother data indeed.) That's only part of the signal spectrum, and this is probably why, surprisingly, DUACS better matches the drifters than GLORYS. I think, results from Figure 1 and Figure 2 need a deeper analysis and comment in **§3.2**.
**Response:** Figure 3 does not directly compare WOC-NATL3D velocities to DUACS velocities, as suggested by the reviewer. Instead, Figure 3 illustrates the difference between the RMSD computed for DUACS and the drifters, and the RMSD computed for WOC-NATL3D and the drifters. To clarify, this is the difference between Figure 2d and Figure 2b. The red color in the figure indicates that WOC-NATL3D displays currents that are closer to the drifters compared to other products. Consequently, Figure 3 highlights the enhancement of our product with respect to both satellite data and GLORYS.
The reviewer raises the issue of temporal smoothing in DUACS, suggesting that it may result in better matches with drifters compared to GLORYS, even though DUACS exclusively represents geostrophic currents. As mentioned in the manuscript line 240, however, GLORYS has also undergone a 5-day smoothing process to maintain consistency across all datasets. The reviewer seems to expect RMSDs lower with GLORYS than DUACS. In fact, while the physics might be better solved in the model, discrepancies in the location of mesoscale features might instead lead to errors in comparison with drifters (actually suffering also from double penalty effects).

**14) Lines 246 to 257: Figure 4 and 15m statistics**: it could be interesting to also add DUACS in the 15m comparison. Can you explain why bias and RMSD are lower at 15m compared to the surface? It could have been interesting to provide statistics on zonal and meridional components separately to assess the Ekman spiral from the surface to 15m depth.
**Response:** The occurrence of larger biases at the surface than at 15 m is expected, as Ekman currents are much stronger at the surface, so that uncertainties on that component weight more there than at 15 m.
As suggested, we plotted the DUACS statistics at 15 m below. Nonetheless, we decided not to include it in the paper because the geostrophy does not change much between the surface and 15 m depth.

[Figure]

[Figure]

**15) Lines 258 to 270**: validation at 1000m depth. Can you provide the mean bias value of SODA? From **Figure 6**, it seems that SODA provide the lower RMSD and bias of the 3 simulations. This is not commented here. Can you complete this analysis?

**Response:** Thanks for the remark. The SODA bias is -1.3 cm s⁻¹ and the value has now been added to line 297: "[...]and with a negative mean bias of 1.3 cm s−1, substantially lower than the other products."

We agree with the reviewer. At 1000 m SODA seems to provide better results than both GLORYS and WOC-NATL3D and we already mentioned in the manuscript that WOC-NATL3D displays the highest values of RMSD of all (l. 298-299).

**16) Lines 284 to 285**: once again, the authors only discussed on current intensity. So, can it be relevant for vertical velocity assessment?

**Response**: We understand the reviewer's point and we have incorporated the new figure, presented below, illustrating the differences between WOC-NATL3D RMSDs and OMEGA3D RMSDs of of the horizontal velocity individual components (u and v), at the surface. In the manuscript, the reviewer should refer to figure 13. It predominantly shows positive values, signifying that WOC-NATL3D horizontal velocities align more closely with in-situ velocities, both in magnitude and direction, compared to OMEGA3D. Some text has been modified to incorporate this new information (l.. 321-324): "Differences between WOC-NATL3D RMSDs and OMEGA3D RMSDs of the horizontal velocity intensity and of its directional components at the surface are illustrated in Fig. 12 and Fig. 13. It reveals essentially positive values, which indicates that WOC-NATL3D horizontal velocities are closer to in-situ velocities than OMEGA3D horizontal velocities, both in magnitude and direction."

[Figure]

[Figure]

Figure 13. Difference between WOC-NATL3D RMSD and OMEGA3D RMSD zonal component (a) and meridional component (b) at the surface. The RMSDs are computed from GDP drifter velocities, over the period 2010-2016.

**17) 3.3 Spectral analysis:** conclusion of this section is that the effective resolution of WOC-NATL3D in the Gulf Stream area is near 1⁄4°. Can you further comment this result? Is it enough to fit the purpose of the project and the user cases?

**Response:** The WOC project is aimed at  providing observations-based, gap-free data at 1/10° resolution, although the foreseen effective-resolution target was declared around 30 km and 3 days. Achieving the nominal 1/10° resolution proves to be quite challenging, particularly for the specific case of the WOC-NATL3D (and 2D) product.

In the reconstruction algorithm for 3D currents, the input data used are themselves gap-free, observations-based fields resulting from interpolating data with gaps. It is known that interpolation has a tendency to smooth the effective spatial-temporal content of L4 data with respect to the nominal one (e.g. Ballarotta et al. 2019).  Nevertheless, reaching a 25 km effective resolution at mid-latitude is considered as a valuable result. Depending on the specific location of our study area, this resolution adequately captures the mesoscale and, in the southernmost section of the basin, a part of sub-mesoscale motions. This is particularly noteworthy given that the Rossby deformation radius spans from 15 to 50 km in the WOC-NATL3D domain (La Casce et al. 2020).

Furthermore, although it falls short of fully recovering the PSD shown by the GLORYS fields (given at 1/12°), WOC-NATL3D product demonstrates a closer agreement with GLORYS currents when compared to SODA and OMEGA3D. This indicates a notable improvement over previous observation-based 3D currents estimates. These mesoscale resolving 3D currents have also already proven successful in describing the impact of ocean dynamics on the marine ecosystem (Munk et al. 2023), justifying the interest of the WOC-NATL3D product for analogous applications.

Ballarotta, M., Ubelmann, C., Pujol, M. I., Taburet, G., Fournier, F., Legeais, J. F., Faugère, Y., Delepoulle, A., Chelton, D., Dibarboure, G., & Picot, N. (2019). On the resolutions of ocean altimetry maps. *Ocean Science*, *15*(4), 1091–1109. https://doi.org/10.5194/OS-15-1091-2019

Lacasce, J. H., & Groeskamp, S. (2020). Baroclinic Modes over Rough Bathymetry and the Surface Deformation Radius. *Journal of Physical Oceanography*, *50*(10), 2835–2847. https://doi.org/10.1175/JPO-D-20-0055.1

Munk, P., Buongiorno Nardelli, B., Mariani, P., & Bendtsen, J. (2023). *Mesoscale-driven dispersion of early life stages of European eel.* https://doi.org/10.3389/fmars.2023.1163125

**18) Conclusion: Lines 328 to 330:** Can the authors say some few words about the differences between background and retrieved horizonal ageostrophic velocity? Here again, validation diagnostics on zonal/meridional components would have been useful for the discussion.
**Response:** The retrieved horizontal ageostrophic velocities are the one obtained once the OMEGA equation is solved for w. They are estimated by integrating two expressions that are obtained during the analytical derivation of the omega equation.

$$u_a(z) = \int_{REF}^{z} \frac{\partial u_a}{\partial z} dz = \frac{1}{f^2} \int_{REF}^{z} \left( \frac{\partial}{\partial x}(N^2 w) - \frac{f}{\rho_0} \frac{\partial}{\partial z}\left(\frac{\partial \tau_{yz}}{\partial z}\right) - 2\frac{g}{\rho_0}\left(\frac{\partial u_g}{\partial x}\frac{\partial \rho}{\partial x} + \frac{\partial v_g}{\partial x}\frac{\partial \rho}{\partial y}\right) - \frac{g}{\rho_0}\frac{\partial}{\partial x}\left(\frac{\partial F_{\rho z}}{\partial z}\right) \right) dz$$

$$v_a(z) = \int_{REF}^{z} \frac{\partial v_a}{\partial z} dz = \frac{1}{f^2} \int_{REF}^{z} \left( \frac{\partial}{\partial y}(N^2 w) + \frac{f}{\rho_0} \frac{\partial}{\partial z}\left(\frac{\partial \tau_{xz}}{\partial z}\right) - 2\frac{g}{\rho_0}\left(\frac{\partial u_g}{\partial y}\frac{\partial \rho}{\partial x} + \frac{\partial v_g}{\partial y}\frac{\partial \rho}{\partial y}\right) - \frac{g}{\rho_0}\frac{\partial}{\partial y}\left(\frac{\partial F_{\rho z}}{\partial z}\right) \right) dz$$

The background ageostrophic velocity is the one approximated by the Ekman currents provided by Copernicus and is used as the reference by further adjusting the viscosity within the Ekman layer to reduce the differences between the background and retrieved horizontal ageostrophic velocities

$$u_{Ekman}(z) = e^{\frac{z}{Damp}}[u_0 \cos(z/D_{rot}) - v_0 \sin(z/D_{rot})]$$

$$v_{Ekman}(z) = e^{\frac{z}{Damp}}[u_0 \sin(z/D_{rot}) + v_0 \cos(z/D_{rot})]$$

where (u0,v0) are the components of the empirical Ekman current at 0 m, and Damp and Drot are the Ekman depth estimates obtained from the amplitude decay and vector rotation between 0 m and 15 m depth.

**Technical comments:**
Repetition of word "computed" in Figure A1 caption. Corrected

---

## Author Comment (AC2)

**Reviewer #2**

The authors present a 1/10° data-driven data set of 3D ocean currents, as well as of temperature and salinity in the upper 1500 meters of the North Atlantic subtropical gyre between 20°N and 50°N, WOC-NATL3D. The data set covers the period from 2010 to 2019 with daily resolution.

The product is based on a diagnostic tool originally developed for a global product (OMEGA3D) by Bruno Buongiorno Nardelli (2022). The method is based on the the quasi-geostrophic omega equation. A deep learning technique is used to obtain the fields from Argo profiles, altimetry, SST and SSS. Also used are ERA5 air-sea fluxes and modelled Ekman currents from Copernicus.

Both products, WOC-NATL3D and OMEGA3D, are supposed to better reproduce drifter observations when compared to reanalysis products. WOC-NATL3D aims to improve accuracy near the surface, in particular by using the modelled Ekman currents. Two reanalysis products (SODA and GLORYS) as well as drifter and altimetry data are used for evaluation.

The article is written well and comprehensibly and also well structured.

We thank the reviewer for their overall comment. Please see below our response to each of your comments. Be aware that the line numbers correspond to the "track-change" version of the manuscript.

**Comments**

**1)** Evaluation of the vertical velocities (section 3.1) is quite limited. I find it understandable that no comparisons with direct measurements are possible. However, an estimation of the uncertainty of the vertical velocities is desirable.

**Response:** Of course we fully agree with the reviewer. As such, we would really be pleased to provide an uncertainty estimation. However, unless the reviewer has some more specific suggestion on the way to obtain it, we are afraid that product intercomparison and indirect assessments are the only viable methods (both already followed in our work).

**2)** I understand that the SODA data set was selected for comparison because vertical velocities are rare in reanalysis products. With GLORYS a second reanalysis product was selected for comparison, is there a justification for this choice?

**Response:** While the SODA dataset provides vertical velocities, its resolution is notably coarse (1/4°). As mentioned in our paper, an alternative method to assess our product is to examine horizontal velocities, which can be inferred from vertical velocities. Our product presents a high horizontal resolution of 1/10°. Therefore, it seems appropriate to compare our product to another one with a reasonably similar resolution. In our view, GLORYS, with a resolution of 1/12°, was a suitable choice for the analysis.

**3)** The labels and titles of the figures are in a small font size. The subscript letters in the titles in particular are difficult to read on a printout.

**Response:** Thank you for bringing this up. We enhanced the font size of labels and titles of all figures.

**l 15**: "On the other way round", I would remove this. Corrected

**l 108**: the surface latent and "sensible" heat flux ? Added (now l. 119)

**l 179**: I can't find an explanation of the meaning of the variable γρ in Eqn. 2. Sorry for the confusion, we meant to write γρ instead of νρ line 185. γρ is a non- local tracer effective gradient. It has been corrected (now l. 195)

**l 182**: I can't find in which equation the variable νρ is used. As we said before, we meant to write γρ instead of νρ.

**l 185**: I would start a new paragraph, before "In order to further improve ...", as the following text focuses on extensions of OMEGA3D. Done (now l. 199)

**l 218**: "... likely due ...", can this be explained further?
**Response:** It is well known that resolution plays a significant role in capturing fine-scale processes and details. However, other factors can contribute to differences in the representation of those processes such as the numerical scheme employed, the choice of vertical coordinate systems, the domain size...
Hence, we used the term "likely due" because we cannot definitively assert that the intensity of vertical velocities is solely attributed to resolution. Multiple factors contribute to the overall behavior, and acknowledging this complexity allows for a more nuanced understanding of the simulation results.

Figs. A1, B1, C1 "computed computed" in the caption. Corrected

---

## Author Comment (AC4)

[revised manuscript text omitted]

**Figure C1.** Number of matchups between WOC-NATL3D, OMEGA3D and GDP drifters at 0 m, computed  over the period 2010-2016 and within 2°x 2° bins.